# Unsafe riding practice among electric bikers in Suzhou, China: an observational study

Jie Yang,[1] Yihe Hu,[2] Wei Du,[3,4] Brent Powis,[5] Joan Ozanne-Smith,[6] Yilan Liao,[7] Ning Li,[3] Ming Wu[1]

JY and YH contributed equally.

For numbered affiliations see end of article.

**Correspondence to**
Dr Ming Wu;
wuming@jscdc.cn

## ABSTRACT

**Background:** Electric bike (E-bike)-related deaths have been increasing rapidly in China and such injuries may be partly attributable to unsafe riding practice.

**Objectives:** To describe potentially unsafe riding behaviours among electric bikers (E-bikers) and to investigate factors influencing these practices in China.

**Methods:** In September 2012, a cross-sectional observation study including a speed measurement component was conducted in Wuzhong (an urban district) and Zhangjiagang (a rural district) of Suzhou, Jiangsu Province, China. Hand-held radar speed metres were used to read travelling speeds of E-bikes and a pro forma observation checklist was used to collect data on road riding practice. Mixed-effect logistic regressions were used to calculate adjusted ORs and 95% CIs for the association between speeding, road rule violations and helmet use and their influencing factors.

**Results:** Among 800 E-bikes with a speed reading, 70.9% exceeded the designed speed limit of 20 km/h. Among a further 20 647 E-bikers observed, 38.3% did not comply with the road rules when entering intersections; and only 2.2% wore helmets. No regional variation was identified between urban and rural areas. Male E-bikers were associated with more speeding and road rule violations, whereas riding a pedal-equipped E-bike was associated with less road rule violations and less helmet use.

**Conclusions:** Unsafe riding practices such as speeding, road rule violations and lack of helmet use were commonplace among E-bikers, especially among men. The study findings indicate that measures aimed at improving E-bike safety are required in China.

## Strengths and limitations of this study

- In this study, we evaluated how fast electric bikers (E-bikers) ride on roads and the possible regional variation of riding behaviours. Furthermore, we investigated factors influencing observed riding behaviours.
- Study findings could provide new evidence to enhance the understanding of on-road riding behaviours among Chinese E-bikers, and to reinforce the imperative to encourage safety gear use and discourage unsafe on-road practices.
- Findings might be limited by lack of generalisability to other settings, possible bias due to unmeasurable confounding and possible misclassification due to measurement errors.

China, more than 120 million E-bikes were registered by 2011[2] and globally an estimated 466 million E-bikes are expected to hit the road by 2016.[3]

In China, bicycle use is shifting to E-bike use. Unfortunately, an associated unwanted shift was observed as E-bike-related deaths increased almost seven times over time from 589 in 2004 to 4029 in 2010 across the nation, whereas bicycle-related deaths decreased three times approximately from 13 655 to 4616 during the same period.[4] Moreover, electric bikers (E-bikers) hospitalised for injuries accounted for 57% of serious non-fatal road traffic injuries and 50% of the direct hospitalisation cost for all road crash casualties in a rural hospital in Suzhou.[5] Thus, E-biker safety is an emerging public health challenge in China.[5–7]

While unsafe riding practices have been reported among E-bikers using a self-reported survey,[2 8] synchronised video camera recording techniques[9] and direct roadside observations,[10] no studies have reported on how fast E-bikers ride on roads allowing for possible regional variation, such as rural/urban disparities, commonly observed for other road safety issues. To enhance the understanding of

## INTRODUCTION

In China, the past few years have witnessed the rapid growth of electric bikes (E-bikes; either with pedals or in scooter form) due to increasing mobility demand when public transportation systems are crowded and inconveniently routed.[1] Growing wealth among Chinese also increases affordability of E-bike purchase, normally priced at approximately US$300. In

on-road riding behaviours among Chinese E-bikers including evaluation of rural/urban variation, we used direct roadside observation techniques to describe their safety practices and hand-held radar metres to estimate their actual travelling speed. We further investigated factors influencing these observed behaviours.

## METHODS

We applied a cross-sectional observation research which comprised two components for this study, that is, observations *with* or *without* speed measurement, which were conducted separately in Suzhou, one of the intervention pilot cities in China for the Bloomberg Philanthropies Global Road Safety Programme (a multinational programme which takes effort to reduce death and serious injury on the roads in 10 low-income and middle-income countries over 5 years extending from 2010 to 2014).[11] Suzhou has the sixth highest gross domestic product per capita on the Chinese mainland, a resident population of 10 million and at least two million E-bikes.[10]

### Field implementation

In Suzhou, two administrative districts, that is, Wuzhong (urban district) and Zhangjiagang (rural district) were selected to conduct both study components. Wuzhong is located in the south of Suzhou metropolitan regions with a resident population of 606 231 in 2012, comprising 49% males and 51% females, whereas Zhangjiagang is located to the north of Suzhou metropolitan regions with a resident population of 909 038 in 2012, comprising 49.2% males and 50.8% females. To select observation sites, a grid was placed over standard maps of Wuzhong and Zhangjiagang, and random digits were generated for each grid box for selection and each valid grid box contained at least one intersection having traffic lights. For each randomly selected site, an alternate site was also selected randomly from the grid as a backup. A pilot study was carried out to validate the field feasibility such as having low volumes of E-bikes for speed measurement; at least two-way motor vehicle lanes, pedestrian crossings and bicycle lanes; enough distance between observation sites so the same E-bikers were unlikely to be observed twice and less likely to interrupt observed behaviours and least likely to increase the crash risk for observers. A total of eight sites (ie, two from each district for each study component) were randomly selected.

For the study component with speed measurement, observers concealed themselves at approximately 50 metres from the corner of the intersection, used hand-held radar speed metres (Bushnell velocity 10-1911CM with measurement range 16–320 km/h)[12] to record the speed metre reading, and collected information on on-road riding behaviours among oncoming E-bikes with valid speed metre reading. For the study component without speed measurement, the observations were conducted at intersection corners. Considering that

traffic characteristics may vary every day, we randomly selected 4 days in a week including one weekend day (September 11, 12, 13 and 15) for the study component with speed measurement and selected another 7-day period (September 17–23) for the study component without speed measurement. The time of day (7:00–18:59) for site observations was set at 2 h intervals as observational periods and randomly assigned to sites.

Four experienced observers were recruited, who had participated in previous roadside E-bike observation studies in other districts in Suzhou.[10] Prior to field implementation, the site observers were trained in specifications of different behaviours, identification of different types of protective items, techniques to observe multiple behaviours especially when an E-bike was moving and data quality control. The site observers formed two groups, that is, the urban group and rural group without rotation. Each group remained in the same district and changed its observational periods and sites every day. Roadside pilot observations and regular on-site audits were conducted to ensure the safety guidelines and accurate and appropriate implementation of the data collection process.

The observers worked in pairs, to observe oncoming E-bikes in ascending distance order. Data items were collected on a pro forma checklist including the type of E-bike (with bike pedals or in scooter form), registration status, rider's gender, couriers or not (in China, many couriers are required to wear uniforms when working and companies provide them uniforms with own logos; although uniforms are in different styles, it' is easy to differentiate couriers from normal E-bikers), carrying passengers, carrying oversized cargo (estimated >60×40×20 cm$^3$, the size of normal airlines carry-on luggage), riding in a motor vehicle lane, running red lights, riding in the opposite direction (ie, facing oncoming traffic), using mobile phone, using helmet, wearing leather gloves and wearing other motorcycle protective clothing. Weather, day of week, time of day, average E-bike traffic volume per minute and presence of a traffic controller (traffic policeman or traffic police assistant directing vehicular and pedestrian traffic) were recorded on separate data collection forms (table 1). Regarding the speed measurement component, for every 10th E-bike, the second observer recorded the radar speed reading during the speed measurement observations.

### Outcome of interest

1. Speeding was defined as binary, that is, yes (travel speed>20 km/h) or no (travel speed≤20 km/h), because E-bikes are manufactured to a mandatory standard[13] with designed maximum travel speed of 20 km/h;
2. Violation was defined as binary, that is, yes (at least one of the following road rule violations was observed: carrying passengers, carrying oversized cargo, riding in a motor vehicle lane, running red lights, riding in the opposite direction or using mobile phone), or no;

**Table 1** Observational item categorisation

| Observational items | Categorisation |
|---|---|
| Weather | Sunny, cloudy or rainy |
| Day of week | Weekday or weekend |
| Time of day | Morning or afternoon |
| Average E-bike traffic volume per minute | Basic (≤10 E-bikes), low (11–15 E-bikes), medium (16–20 E-bikes) or high (>20 E-bikes) |
| Presence of a traffic controller (traffic policeman or traffic police assistant directing vehicular and pedestrian traffic) | Yes or no |
| Type of E-bike | Equipped with pedals or otherwise in scooter form |
| E-bike registration status | Registered with registration plate displayed or otherwise unregistered |
| E-bikers' gender | Male or female |
| E-bikers' occupation | Couriers or not |
| Carrying passengers | Yes or no |
| Carrying oversized cargo (>60×40×20 cm$^3$) | Yes or no |
| Riding in a motor vehicle lane | Yes or no |
| Running red lights | Yes or no |
| Riding in the opposite direction (ie, facing oncoming traffic) | Yes or no |
| Mobile phone use | Yes or no |
| Helmet use | Yes or no |
| Wearing leather gloves | Yes or no |
| Wearing other motorcycle protective clothing | Yes or no |

E-bike, electric bike; E-biker, electric biker.

3. Helmet use was defined as binary, that is, yes (wearing a motorcycle helmet) or no.

## Statistical analysis

Completed observational records were reviewed; and data were entered with double entry. All data analyses were conducted using SAS V.9.2 (SAS Institute, 2002). The interobserver reliability was assessed using κ-statistics and agreement reached at least 85% for each pair of observers. Frequencies and proportions of speed reading and observed on-road riding behaviours were calculated where appropriate. We used mixed logistic regression allowing for random selection of observational sites to estimate OR and associated 95% CI for different study outcomes adjusted for observational items (table 1). Further mixed-effect logistic regression analyses of individual road rule violations (ie, carrying passengers, carrying oversized cargo, riding in a motor vehicle lane, running red lights, riding in the opposite direction or using mobile phone) were stratified by different regions, that is, Wuzhong or Zhangjiagang. We defined p values less than 0.05 as statistically significant.

## RESULTS

There were a total of 27 observational periods (ie, 14 in Wuzhong and 13 in Zhangjiagang) for direct observational data collection on 20 647 E-bikes and 16 periods (ie, 8 for each district) for speed measurement on 800 E-bikes. The average number of E-bikes per observational period was 729 (range 103–1317) and 803 (range 552–1046) for Wuzhong and Zhangjiagang, respectively.

Table 2 describes the observation results and shows that E-bikes were the dominant transportation means in Wuzhong district. Despite similarities across some observational items such as low helmet use (ie, 2.1% in Wuzhong vs 2.2% in Zhangjiagang) and commonplace carrying of passengers (21.3% vs 20.5%), there were differences in riding violations especially for riding licensed E-bikes (96.3% vs 36.3%; table 2).

Approximately, 83.3% (n=333) and 58.5% (n=234) E-bikers were observed travelling at a speed greater than 20 km/h; approximately, 41.3% (n=4211) and 35.4% (n=3700) violating at least one of the listed road rules; and 2.5% (n=251) and 3.1% (n=319) using any safety gear in Wuzhong and Zhangjiagang, respectively.

No statistically significant evidence indicates the existence of regional variation in terms of elevated odds of speeding, general road rule violations or lack of helmet use. Compared with female E-bikers, male E-bikers showed greater ORs of speeding (OR=2.12, 95% CI 1.50 to 3.01) and violation (OR=1.35, 95% CI 1.27 to 1.44). Reduced ORs of road rule violations (OR=0.66, 95% CI 0.62 to 0.70) and wearing a helmet (OR=0.39, 95% CI 0.32 to 0.49) were found to be associated with riding pedal-equipped E-bikes compared with those in scooter form, whereas the highest elevated ORs of helmet use (OR=7.21, 95% CI 4.01 to 12.98) and road rule violations (OR=5.34, 95% CI 3.58 to 7.99) were observed among couriers compared with other E-bikers (table 3).

The results of regional stratification demonstrate that risk factor profile may vary across regions. For example, of riding opposite direction, couriers were associated with significantly elevated OR (2.03, 95% CI 1.03 to 4.00) in Wuzhong but marginally reduced OR (0.71, 95% CI 0.36 to 1.41) in Zhangjiagang, whereas male E-bikers were associated with marginally elevated OR (1.09, 95% CI 0.91 to 1.31) in Wuzhong but significantly reduced OR (0.89, 95% CI 0.81 to 0.97) in Zhangjiagang (tables 4 and 5).

## DISCUSSION

Poor safety practice was commonplace including speeding, road rule violations and little use of helmets and this did not vary between rural and urban areas. Male E-bikers seemed to bear more risks of speeding and road rule violations. Although couriers were seven times more likely to wear a helmet when riding an E-bike, they were also five times more likely to violate road rules when entering an intersection compared with other E-bikers. When riding E-bikes with pedals rather than

**Table 2** Distribution of observational items among E-bike study populations

| Sample size | (n) | Wuzhong (urban district) | | Zhangjiagang (rural district) | |
| | | With speed measure 400 | Without speed measure 10 202 | With speed measure 400 | Without speed measure 10 445 |
| --- | --- | --- | --- | --- | --- |
| Traffic mix | E-bikes (%) | 44.6 | 46.5 | 28.2 | 34.5 |
| | Pedestrians (%) | 7.4 | 9.1 | 3.0 | 10.6 |
| | Bicycles (%) | 3.5 | 3.2 | 9.6 | 4.7 |
| | Cars (%) | 36.2 | 35.1 | 57.1 | 47.6 |
| | Heavy vehicles (%) | 8.3 | 6.2 | 2.2 | 2.7 |
| E-bike volume per minute (basic as ≤10 E-bikes, | Basic (%) | 25.0 | 40.9 | 50.0 | 40.2 |
| low as 11–15 E-bikes, medium as16–20 E-bikes | Low (%) | 12.5 | 17.0 | 37.5 | 59.8 |
| or high as >20 E-bikes) | Medium (%) | 37.5 | 10.4 | 12.5 | 0.0 |
| | High (%) | 25.0 | 31.7 | 0.0 | 0.0 |
| Weather | Sunny (%) | 75.0 | 82.8 | 100.0 | 69.8 |
| | Cloudy (%) | 0.0 | 9.4 | 0.0 | 30.2 |
| | Rainy (%) | 25.0 | 7.8 | 0.0 | 0.0 |
| Day of week | Weekday (%) | 75.0 | 75.7 | 75.0 | 70.5 |
| Time of day | Morning (%) | 37.5 | 45.7 | 50.0 | 39.5 |
| Traffic controller | Yes (%) | NA | 0.0 | NA | 25.4 |
| Occupation | Courier (%) | 1.8 | 0.8 | 1.3 | 0.5 |
| Gender | Males (%) | 69.0 | 59.8 | 58.5 | 51.7 |
| Registration | Yes (%) | 94.0 | 96.3 | 35.3 | 36.3 |
| Pedals | Yes (%) | 28.0 | 37.1 | 53.3 | 54.3 |
| Carrying passengers | Yes (%) | 24.3 | 21.3 | 11.0 | 20.5 |
| Carrying large cargo | Yes (%) | 11.5 | 6.9 | 15.8 | 10.1 |
| Riding in a motor vehicle lane | Yes (%) | 13.5 | 3.1 | 2.3 | 1.4 |
| Riding opposite direction | Yes (%) | 3.0 | 5.7 | 25.0 | 30.8 |
| Mobile phone use | Yes (%) | 0.8 | 0.7 | 2.0 | 1.1 |
| Helmet use | Yes (%) | 3.3 | 2.1 | 5.0 | 2.2 |
| Gloves | Yes (%) | 0.8 | 0.4 | 1.8 | 0.8 |
| Running red lights | Yes (%) | NA | 16.5 | NA | 6.2 |

E-bike, electric bike.

those in scooter form, E-bikers had a lower likelihood of violating road rules and wearing a helmet. These identified safety gaps build on previous evidence[5–10] identifying the need to discourage unsafe practice and encourage safety gear use among E-bikers in China, particularly in the context of China recently joining global action to improve road safety in the next decade.[11]

Consistent with previous studies,[2 8–10] this study confirmed a range of factors associated with observed E-biker behaviours and revealed the invariant nature of unsafe E-bike riding practice in general. We conducted a similar study during March 2012 in metropolitan Suzhou areas and found 27% of E-bikers violated at least one road rule and 41% used at least one type of safety gear.[10] The current study identified a somewhat higher prevalence of road rule violations (38%) and lower safety gear use (3%). This variation may be explained by the seasonality, for example, the sharp drop in glove use (from 37% to 0.6%) and helmet use (from 9% to 2%). Zhang et al[14] reported a similar decrease in helmet use among

motorcycles in Guangxi during the hot and humid season. The study findings also relate to the previous reports of increasing E-biker deaths and injuries across Mainland China,[4–7] which echoes the call for action to develop policies to improve E-bike safety in China.

On the basis of the study findings, measures aimed at improving E-bike safety are required. For example, the observed high prevalence of unsafe riding practices implies a need for policy change. Current road rules regulate E-bikes as pedal bicycles that should travel in non-motor vehicle lanes at a maximum speed of 15 km/h,[15] whereas the mandatory standard for E-bikes specifies a maximum speed of 20 km/h and a maximum weight of 40 kg in addition to requiring a specified braking distance and pedal installment.[13] However, these specifications may not be widely enforced as most of the electric two-wheelers are not designed and produced in line with the national standards of non-motor vehicles (Tsinghua Law School. *Aiming at behavioral changes: improving drink driving and speeding law in China*. Beijing: Tsinghua

**Table 3** Adjusted ORs (95% CI) for speeding, road rule violations and helmet use among electric bikers*

| | Speeding N=800 | Violations N=20 647 | Helmet use N=20 647 |
|---|---|---|---|
| **Region** | | | |
| Urban | 1.14 (0.66 to 1.99) | 1.01 (0.95 to 1.07) | 0.98 (0.84 to 1.14) |
| Rural | | Reference | |
| **Weather** | | | |
| Sunny | 0.29 (0.02 to 3.58) | 0.89 (0.40 to 1.98) | 0.58 (0.19 to 1. 80) |
| Cloudy | – | 0.76 (0.31 to 1.88) | 0.55 (0.16 to 1.98) |
| Rainy | | Reference | |
| **Weekday** | | | |
| Yes | 1.73 (0.92 to 3.25) | 0.86 (0.53 to 1.42) | 1.43 (0.72 to 2.81) |
| No | | Reference | |
| **Time of day** | | | |
| Morning | 0.73 (0.47 to 1.15) | 1.07 (0.69 to 1.65) | 1.01 (0.56 to 1.82) |
| Afternoon | | Reference | |
| **Volume** | | | |
| Basic | 0.27 (0.02 to 3.24) | 1.53 (0.77 to 3.02) | 0.86 (0.35 to 2.14) |
| Low | 0.50 (0.04 to 6.11) | 1.29 (0.62 to 2.67) | 0.90 (0.34 to 2.37) |
| Medium | 0.16 (0.01 to 2.06) | 1.13 (0.36 to 3.60) | 0.66 (0.14 to 3.02) |
| High | | Reference | |
| **Traffic control** | | | |
| Yes | – | 0.76 (0.41 to 1.38) | 1.57 (0.70 to 3.51) |
| No | | Reference | |
| **Gender** | | | |
| Male | *2.12 (1.50 to 3.01)* | *1.35 (1.27 to 1.44)* | 0.66 (0.54 to 0.80) |
| Female | | Reference | |
| **Courier** | | | |
| Yes | 0.75 (0.20 to 2.82) | *5.34 (3.58 to 7.99)* | *7.21 (4.01 to 12.98)* |
| No | | Reference | |
| **Registration** | | | |
| Yes | 0.96 (0.63 to 1.44) | *0.82 (0.75 to 0.88)* | 1.18 (0.92 to 1.52) |
| No | | Reference | |
| **Pedals** | | | |
| Yes | 0.79 (0.56 to 1.12) | *0.66 (0.62 to 0.70)* | *0.39 (0.32 to 0.49)* |
| No | | Reference | |

Significant results are highlighted in italics.
*The adjusting variables were observational items shown in table 1.

University, unpublished report, 2012). With regard to those that do comply with the national standards of non-motor vehicles, producers, for marketing purposes, often install the so called 'speed limiting devices' on their products. With the speed-limiting devices, the maximum speed by which the electric two-wheelers could operate is 20 km/h as required by the mandatory standard for E-bikes, whereas the speed-limiting devices are designed and installed in a way that could be easily dismantled by customers themselves or by sales persons. Without speed-limiting devices, the speed of these electric two-wheelers could effortlessly go beyond 20 km/h and may be up to 40 km/h.[16]

Notably, the *Safety Specifications for Power Driven Vehicles Operating on Roads* defines a motorcycle as being power-driven with the maximum speed exceeding 50 km/h and a moped with a maximum speed range from 20 to 50 km/h.[17] This has the legal implication that any E-bikes (mostly in scooter form) that could travel faster than 20 km/h should be regulated as motor vehicles by the road rules. Obviously, such conflict between the mandatory standard for E-bikes and road rules might create difficulties for legislative enforcement, including cities where motorcycles are banned.

In addition, the low use of helmets also implies a need for policy change given that the effectiveness of helmets in head injury prevention is well established for bicyclists[18] and motorcyclists.[19] Regardless of the introduction of compulsory motorcycle helmet use producing a substantial increase in using of helmets among motorcyclists in China,[20] similar regulations were missing for bicyclists and E-bikers. Therefore, road rule revisions to encourage helmet use among E-bikers are urgently needed and should be incorporated into the broad road safety agenda.

To the best of our knowledge, no international E-bikers' riding practice studies were conducted before. Compared with other cross-sectional observational

**Table 4** Adjusted ORs (95% CI) for individual road rule violations among electric bikers observed in Wuzhong (urban district)*

| | Carrying passengers N=2169 | Running red lights N=1682 | Carrying large cargo N=700 | Riding opposite direction N=579 | Riding in a motor vehicle lane N=315 | Mobile phone use N=72 |
|---|---|---|---|---|---|---|
| **Weather** | | | | | | |
| Sunny | 2.51 (0.64 to 9.86) | 0.41 (0.06 to 2.58) | 1.16 (0.62 to 2.18) | 1.54 (0.49 to 4.84) | 1.03 (0.07 to 15.00) | 0.80 (0.16 to 4.05) |
| Cloudy | 2.01 (0.32 to 12.74) | 0.78 (0.07 to 9.51) | 1.28 (0.56 to 2.93) | 1.43 (0.31 to 6.62) | 7.72 (0.21 to 288.20) | 3.17 (0.34 to 29.69) |
| Rainy | | | Reference | | | |
| **Weekday** | | | | | | |
| Yes | 0.55 (0.16 to 1.92) | 1.21 (0.22 to 6.62) | 1.15 (0.64 to 2.06) | 0.83 (0.29 to 2.36) | 0.45 (0.04 to 5.31) | 0.98 (0.14 to 6.69) |
| No | | | Reference | | | |
| **Time of day** | | | | | | |
| Morning | 0.71 (0.22 to 2.33) | 2.35 (0.47 to 11.70) | 1.22 (0.70 to 2.10) | 0.90 (0.34 to 2.42) | 0.86 (0.08 to 8.97) | 1.60 (0.32 to 8.00) |
| Afternoon | | | Reference | | | |
| **Volume** | | | | | | |
| Basic | 0.53 (0.15 to 1.92) | 9.21 (1.63 to 51.92) | *3.59 (2.01 to 6.41)* | 0.90 (0.31 to 2.59) | *47.54 (3.69 to 612.34)* | 5.15 (1.13 to 23.42) |
| Low | 0.65 (0.11 to 4.01) | 4.95 (0.43 to 57.61) | *2.65 (1.12 to 6.25)* | 0.86 (0.19 to 3.98) | 3.86 (0.11 to 133.67) | 3.51 (0.32 to 37.89) |
| Medium | 0.43 (0.07 to 2.76) | 8.41 (0.68 to 104.11) | *1.88 (0.81 to 4.36)* | 0.83 (0.18 to 3.89) | 7.59 (0.20 to 296.13) | 4.67 (0.43 to 50.91) |
| High | | | Reference | | | |
| **Gender** | | | | | | |
| Male | *1.26 (1.13 to 1.40)* | *1.32 (1.17 to 1.49)* | *1.73 (1.43 to 2.10)* | 1.09 (0.91 to 1.31) | *2.14 (1.62 to 2.84)* | *2.38 (1.29 to 4.40)* |
| Female | | | Reference | | | |
| **Courier** | | | | | | |
| Yes | *0.11 (0.03 to 0.43)* | 0.82 (0.44 to 1.55) | *53.36 (31.34 to 90.86)* | *2.03 (1.03 to 4.00)* | 1.14 (0.34 to 3.76) | *5.26 (1.79 to 15.46)* |
| No | | | Reference | | | |
| **Registrations** | | | | | | |
| Yes | 1.17 (0.89 to 1.54) | 0.99 (0.75 to 1.32) | *0.69 (0.48 to 0.99)* | 1.07 (0.68 to 1.66) | 0.74 (0.45 to 1.23) | 3.16 (0.43 to 22.94) |
| No | | | Reference | | | |
| **Pedals** | | | | | | |
| Yes | *0.69 (0.62 to 0.77)* | 0.89 (0.79 to 1.01) | *0.62 (0.51 to 0.75)* | 0.99 (0.83 to 1.20) | *0.57 (0.43 to 0.77)* | 0.82 (0.47 to 1.42) |
| No | | | Reference | | | |

*ORs were adjusted for observational items in table 1 (significant results are highlighted in italics).

**Table 5** Adjusted ORs (95% CI) for individual road rule violations among electric bikers observed in Zhangjiagang (rural district)*

| | Riding opposite direction N=3220 | Carrying passengers N=2139 | Carrying large cargo N=1058 | Running red lights N=648 | Riding in a motor vehicle lane N=141 | Mobile phone use N=116 |
|---|---|---|---|---|---|---|
| **Weather** | | | | | | |
| Sunny | 0.75 (0.54 to 1.05) | 0.93 (0.57 to 1.53) | 0.92 (0.54 to 1.55) | 0.14 (0.01 to 2.19) | 0.64 (0.09 to 4.39) | 1.35 (0.05 to 37.20) |
| Cloudy | | | Reference | | | |
| **Weekday** | | | | | | |
| Yes | 0.78 (0.56 to 1.08) | *0.46 (0.28 to 0.75)* | *0.53 (0.31 to 0.88)* | 1.86 (0.12 to 28.19) | 1.91 (0.32 to 11.50) | 1.16 (0.04 to 36.12) |
| No | | | Reference | | | |
| **Time of day** | | | | | | |
| Morning | 1.15 (0.90 to 1.48) | 0.86 (0.59 to 1.25) | 0.87 (0.59 to 1.30) | 4.83 (0.58 to 40.31) | 0.86 (0.19 to 3.99) | 0.80 (0.06 to 10.56) |
| Afternoon | | | Reference | | | |
| **Volume** | | | | | | |
| Basic | 1.15 (0.88 to 1.51) | 1.04 (0.69 to 1.56) | 0.97 (0.63 to 1.49) | 0.11 (0.01 to 1.11) | 0.35 (0.08 to 1.62) | 2.55 (0.14 to 47.09) |
| Low | | | Reference | | | |
| **Traffic control** | | | | | | |
| Yes | *1.86 (1.36 to 2.54)* | 1.33 (0.84 to 2.12) | *1.97 (1.20 to 3.21)* | *0.03 (0.00 to 0.39)* | *0.05 (0.01 to 0.40)* | 1.26 (0.06 to 25.40) |
| No | | | Reference | | | |
| **Gender** | | | | | | |
| Male | *0.89 (0.81 to 0.97)* | *1.17 (1.05 to 1.29)* | 1.14 (0.99 to 1.30) | *1.32 (1.11 to 1.57)* | 1.13 (0.80 to 1.60) | 1.75 (0.14 to 22.54) |
| Female | | | Reference | | | |
| **Courier** | | | | | | |
| Yes | 0.71 (0.36 to 1.41) | Not estimable | *11.06 (6.05 to 20.23)* | 1.07 (0.37 to 3.12) | 2.67 (0.62 to 11.53) | Not estimable |
| No | | | Reference | | | |
| **Registrations** | | | | | | |
| Yes | *0.86 (0.78 to 0.93)* | *0.76 (0.68 to 0.84)* | *0.87 (0.76 to 0.99)* | 1.03 (0.86 to 1.22) | *0.59 (0.40 to 0.88)* | 0.98 (0.08 to 11.86) |
| No | | | Reference | | | |
| **Pedals** | | | | | | |
| Yes | 0.92 (0.84 to 1.00) | *0.78 (0.71 to 0.87)* | *0.44 (0.38 to 0.50)* | 0.91 (0.77 to 1.08) | 0.95 (0.67 to 1.34) | 0.99 (0.09 to 11.53) |
| No | | | Reference | | | |

*ORs were adjusted for observational items in table 1 (significant results are highlighted in italics).

studies, this study is limited by a lack of generalisability to other settings (different regions); possible bias due to unmeasurable confounding (influence of road infrastructure) and possible misclassification due to measurement errors (incorrect speed reading). Thus, care should be taken when interpreting the study findings. To minimise the likelihood of measurement errors, various small-scale pilot studies were conducted to determine the feasibility of the study and to validate the observational instruments. Moreover, this study established a strict quality control scheme and recruited experienced observers who had participated in previous studies using similar techniques.[10] Thus, misclassification may not bias the key findings to an important degree. Nevertheless, the study findings provide new evidence to complement previous findings as to diverse safety issues among E-bikers, and to reinforce the imperative to encourage safety gear use and discourage unsafe on-road practices.

## CONCLUSION

E-bikes are becoming a dominant road transportation means for commuters in China, and they are increasingly used as a sustainable alternative to traditional transportation in other countries because of the low maintenance cost and low polluting mobility. The observed unsafe riding practices signal emerging road safety challenges in China and in similar settings elsewhere. Translating established safety practices, such as helmet use, and enforcing existing countermeasures, such as speed limit devices, may be used to improve safety practice among E-bikers. A strong political will is especially needed to leapfrog substantial losses associated with E-bike risk in China without sacrificing mobility needs.

**Author affiliations**
[1]Department of Chronic Disease Control, Jiangsu Provincial Centre for Disease Control and Prevention, China
[2]Department of Chronic Disease Control, Suzhou Municipal Centre for Disease Control and Prevention, Suzhou, China
[3]Institute of Population Research, Peking University, China
[4]Neuroscience Research Australia, University of New South Wales, Australia
[5]School of Postgraduate Studies and Research, International Medical University, Malaysia
[6]Department of Forensic Medicine, Monash University, Australia
[7]Institute of Geographical Sciences and Nature Resources Research, Chinese Academy of Sciences, China

**Acknowledgements** The authors would like to appreciate the special support provided by the WHO China Office, the Ministry of Health in China, Suzhou Bureau of Health, and Suzhou Bureau of Public Security. The authors would also like to thank Ziyi Jin, Jianfeng Liu, Qi Zhang, Xianglin Liu and Yan Lu for their assistance in field work.

**Contributors** JY and YH contributed equally to the study design, research implementation, literature review, data analysis, fully wrote the first draft and contributed to subsequent drafts. All the other authors contributed to the conceptual development, data interpretation, critical revision of the first draft of the manuscript and subsequent drafts.

**Funding** This work was funded by the Bloomberg Philanthropies Global Road Safety Programme (WHO Registration 2011/199579-0), Jiangsu Provincial

Centre for Disease Control and Prevention (JKRC2011014), National Natural Science Foundation of China (No. 41101431).

**Competing interests** WD was supported by an NHMRC fellowship. YL was supported by the National Natural Science Foundation of China (No. 41101431).

**Ethics approval** Jiangsu Provincial Centre for Disease Control and Prevention.

**Provenance and peer review** Not commissioned; externally peer reviewed.

**Data sharing statement** No additional data are available.

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
