## [Reviewer comments · BMJ Open]

Some articles will have been accepted based in part or entirely on reviews undertaken for other BMJ Group journals. These will be reproduced where possible.

ARTICLE DETAILS

TITLE (PROVISIONAL)	Unsafe riding practice among electric bikers in Suzhou, China: an observational study
AUTHORS	Yang, Jie; Hu, Yihe; Du, Wei; Powis, Brent; Ozanne-Smith, Joan; Liao, Yilan; Li, Ning; Wu, Ming

VERSION 1 - REVIEW

REVIEWER	Elaheh Ainy Safety Promotion and Injury Prevention Research Center of Shahid Beheshti University of Medical Sciences, Tehran, Iran
REVIEW RETURNED	19-Sep-2013

GENERAL COMMENTS	Abstract: need to be revised and completed.
---

REVIEWER	Carlos Martin Cantera Unitat Suport Recerca Barcelona. IDIAP Jordi Gol Department of Medicine. Universitat Autònoma Barcelona
REVIEW RETURNED	18-Oct-2013

GENERAL COMMENTS	Comment 1: Introduction Could the authors provide the rise in the number of E-bikers for the period 2004-2010? (Page 5, second paragraph, line 1). Comment 2: Introduction In the second paragraph it may be easier to understand the magnitude of E-biker and bicycle related fatalities if the authors specify that the first one has increased almost 7 times and the second one has decreased 3 times approximately (Page 4, second paragraph, lines 3 and 4) Comment 3: Methods Could the authors state the type of study in the first sentence (as in the abstract) (Page 5)
--

Comment 4: Methods

Could the authors explain briefly, in a sentence, the Bloomberg Philantropies Global Road Safety Programme? (Page 5, first paragraph of the Methods section).

Comment 5: Methods

Do the authors have any reference for the radar used? If yes please state it. (Page 6, first paragraph, line 3).

Comment 6: Methods

When the authors explain when the observations were done, do they mean that for those observations where velocity was collected by radar, speed record and characteristics were collected in different days? If this is true could they explain why? (Page 6, first paragraph, lines 6, 7 and 8).

Comment 7: Methods

Were the observers always in the same place (urban or rural) or did they rotate and the same observer was one day in the rural area and the other day in the urban one? (Page 6, second paragraph).

Comment 8: Methods

Please introduce the word 'uniformed' when talking about couriers (Page 6, third paragraph, line 4).

Comment 9: Results

As seen in table 2, many observational items are quite different among urban and rural setting, could the authors provide some sociodemographic characteristics of both regions e.g: percentage of males and females, age distribution).

Comment 10: Results

We were not able to find the percentages 3.7% and 63.7% in table 2.

Comment 11: Results

Please, in the paragraph where results of the logistic regression are offered, could the authors provide some values of the ORs in the text? Why is not the highest OR (7.21) reported on the text since is the highest one (but is mentioned on the discussion section)?

Comment 12: Results

Since table 2 shows differences regarding observational items among urban and rural areas, it may be useful to undertake the logistic regression separately in the two districts.

Comment 13: Discussion

Is there any international study (not undertaken in China) that can be used to compare their results? If not, please indicate in the text something like: 'To our best knowledge, no international studies....'

Comment 14: Table 2

Could the authors specify that Wuzhong district is the urban one and Zhangjiagang is the rural in the heading of the table in order to make it more understandable for the readers that are not familiar with Chinese districts?

Comment 15: Table 2

What does 'n/a' stands for? Could the authors specify it?

Comment 16: Table 3

Could the authors specify the adjusting variables?

Comment 17: Table 3

We have assumed that the Ors for the column 'speeding' has been calculated with the 800 speeding observations, but could the authors specify it on the table? The OR for the 'violations' and 'safety gear use' columns were calculated with the 800 speeding observations or with the 20,647 direct observations. Please specify.

VERSION 1 – AUTHOR RESPONSE

Reviewer: Elaheh Ainy

Q2-1: Please state any competing interests or state 'None declared': There is no conflict of interest.

A2-1: As to the reviewer's suggestion, we revised the declaration as "There is no conflict of interest" (Page15, line3) .

Q2-2: Abstract: need to be revised and completed.

A2-2: We appreciate the reviewer's suggestion and have rephrased the abstract. (Page 2) .

Reviewer: Carlos Martin Cantera

Q3-1: Please state any competing interests or state 'None declared': None declared

A3-1: As to the reviewer's suggestion, we have revised it. (Page15, line3)

Q3-2 Comment 1: Introduction

Could the authors provide the rise in the number of E-bikers for the period 2004-2010? (Page 5, second paragraph, line 1).

A3-2: We tried to find relevant information but did not find it. In China, health and public security departments are responsible for collecting casualty number of E-bikers, while the E-bike association collects vehicle numbers. However, such statistics were not publicized regularly.

Q3-3 Comment 2: Introduction

In the second paragraph it may be easier to understand the magnitude of E-biker and bicycle related fatalities if the authors specify that the first one has increased almost 7 times and the second one has decreased 3 times approximately (Page 4, second paragraph, lines 3 and 4)

A3-3: We appreciate the reviewer's suggestion and revised it as 'Unfortunately, an associated unwanted shift was observed as E-bike related fatalities increased almost 7 times over time from 589 in 2004 to 4,029 in 2010 across the nation, whereas bicycle related fatalities decreased 3 times approximately from 13,655 to 4616 during the same period' in our paper. (Page4 second paragraph, line2 and 4)

Q3-4 Comment 3: Methods

Could the authors state the type of study in the first sentence (as in the abstract) (Page 5)

A3-4: We thank for the reviewer's reminder and state study type in the Methods as 'We applied a cross-sectional observation research which comprised two components for this study' (Page 5, Methods, line 1 and 2).

Q3-5 Comment 4: Methods

Could the authors explain briefly, in a sentence, the Bloomberg Philantropies Global Road Safety Programme? (Page 5, first paragraph of the Methods section).

A3-5: The Bloomberg Philantropies Global Road Safety Programme is a multinational programme which takes effort to reduce death and serious injury on the roads in ten low-and middle-income countries from 2010 to 2014.As suggested by the reviewer, we add this sentence in the manuscript (Page 5, Methods paragraph line 4 to 6) .

'This is a multinational programme which take effort to reduce death and serious injury on the roads in ten low-and middle-income countries over five years extending from 2010 to 2014'.

Q3-6 Comment 5: Methods

Do the authors have any reference for the radar used? If yes please state it. (Page 6, first paragraph, line 3).

A3-6: We thank for the review's suggestion. Bushnell Velocity Radar Gun 101911 is a kind of easy-to-use, point-and-shoot pistol grip. It can display fastest speed once trigger is released and measure speeds from 10-200 MPG/16-177 KPH, ideal for all sports and vehicles. We add the introduction of this radar as a reference in our manuscript. <http://www.bushnell.com/all-products/outdoor-technology/velocity-speed-gun>

Q3-7 Comment 6: Methods

When the authors explain when the observations were done, do they mean that for those observations where velocity was collected by radar, speed record and characteristics were collected in different days? If this is true could they explain why? (Page 6, first paragraph, lines 6, 7 and 8).

A3-7: As the reviewer mentioned, in this study, different days were selected for observation and speed collection. The reason is, there might be difference in the traffic between working days and weekends, therefore, we randomly selected 5 working days and 2 days in weekend for observation and 4 working days and 1 weekend day for speed collection to improve the reliability.

Q3-8 Comment 7: Methods

Were the observers always in the same place (urban or rural) or did they rotate and the same observer was one day in the rural area and the other day in the urban one? (Page 6, second paragraph).

A3-8: The trained observers worked in pairs. One group worked in urban district (Wuzhong) while the other group worked in rural district (Zhangjiagang), they didn't rotate. Both groups changed their observational periods and sites every day within one place.

Q3-9 Comment 8: Methods

Please introduce the word 'uniformed' when talking about couriers (Page 6, third paragraph, line 4).

A3-9: In China, many couriers are required to wear uniforms when working and companies provide them uniforms with own logos; although uniforms are in different styles, it's easy to differentiate couriers from normal E-bikers. We marked it in the paper. (Page 6 third paragraph line 3-5)

Q3-10 Comment 9: Results

As seen in table 2, many observational items are quite different among urban and rural setting, could the authors provide some sociodemographic characteristics of both regions e.g: percentage of males and females, age distribution).

A3-10: WuZhong was an urban district located in the south of Suzhou metropolis. It had a resident population of 606,231 for which male and female accounted 49.0% , 51.0% respectively in 2012.

While Zhangjiagang which located in the north of Suzhou was an rural area with a resident population of 909,038 in 2012, male and female accounted for 49.2%, 50.8% respectively.

Q3-11 Comment 10: Results

We were not able to find the percentages 3.7% and 63.7% in table 2.

A3-11: The percentages 3.7% and 63.7% should be the proportion of unlicensed E-bikes. We appreciate reviewer's carefulness and we have changed the description. (Page 9, line 3 and 4)

Q3-12 Comment 11: Results

Please, in the paragraph where results of the logistic regression are offered, could the authors provide some values of the ORs in the text? Why is not the highest OR (7.21) reported on the text since is the highest one (but is mentioned on the discussion section)?

A3-12: As suggested by the reviewer, we rephrased that paragraph as follow 'Compared with female E-bikers, males showed greater ORs of speeding (OR=2.12, 95%CI=1.50-3.01) and violation (OR=1.35, 95%CI=1.27-1.44). Reduced ORs of road rule violations (OR=0.66, 95%CI=0.62-0.70) and wearing a helmet (OR=0.39, 95%CI=0.32-0.49) were found to be associated with riding pedal-equipped E-bikes compared with those in scooter form; whereas the highest elevated ORs of Helmet use (OR=7.21, 95%CI=4.01-12.98) and road rule violations (OR=5.34, 95%CI=3.58-7.99) were observed among couriers compared with other E-bikers' (Page 10, first paragraph line 3-9)

Q3-13 Comment 12: Results

Since table 2 shows differences regarding observational items among urban and rural areas, it may be useful to undertake the logistic regression separately in the two districts.

A3-13: We thank and totally agree with the reviewer's comment. The differences regarding observational items between urban and rural areas do exist, due to the limitation of page, we did not present the results of logistic regression by different areas and we have planned to display it in next paper

Q3-14 Comment 13: Discussion

Is there any international study (not undertaken in China) that can be used to compare their results? If not, please indicate in the text something like: 'To our best knowledge, no international studies....'

A3-14: As far as we know, there isn't any international observational study about E-bikers riding practice has been published. Thus, we rephrased our discussion according to reviewer's suggestion. (Page 13, Third paragraph, line 1 and 2)

'To our best knowledge, no international E-bikers' riding practice studies were conducted before. Compared with other cross-sectional observational studies....'

Q3-15 Comment 14: Table 2

Could the authors specify that Wuzhong district is the urban one and Zhangjiagang is the rural in the heading of the table in order to make it more understandable for the readers that are not familiar with Chinese districts?

A3-15 : We thank for this suggestion and have revised in the Table2 (Page 9) .

Q3-16 Comment 15: Table 2

What does 'n/a' stands for? Could the authors specify it?

A3-16: 'n/a' stands for 'Not Applicable'. In Table2 we used it in variables 'traffic controller' and 'running red lights' because the speed measure site should be no 'traffic controller' and away from 'running red lights'. We have specified it in the table.

Q3-17Comment 16: Table 3

Could the authors specify the adjusting variables?

A3-17: The adjusting variables of Table3 were observational items showed in Table1, We added a footnote in Table3 (Page 10) .

Q3-18 Comment 17: Table 3

We have assumed that the ORs for the column 'speeding' has been calculated with the 800 speeding observations, but could the authors specify it on the table? The OR for the 'violations' and 'safety gear use' columns were calculated with the 800 speeding observations or with the 20,647 direct observations. Please specify.

A3-18: We appreciate the reviewer's reminder. For the column 'speeding', it has been calculated with the 800 observations, The OR for the 'violations' and 'helmet use' columns were calculated with the 20,647 direct observations. We have addressed the sample sizes separately in Table3 (page 10).